# Reduced mood variability is associated with enhanced performance during ultrarunnning

**Paul Burgum, Daniel T. Smith**[ID]*

Department of Psychology, Durham University, Durham, United Kingdom

* daniel.smith2@durham.ac.uk

## Abstract

Ultrarunning requires extraordinary endurance but the psychological factors involved in successful ultrarunning are not well understood. One widely held view is that fluctuations in mood play a pivotal role in performance during endurance events. However, this view is primarily based on comparisons of mood before and after marathons and shorter running events. Indeed, to date no study has explicitly examined mood changes *during* a competive ultramarathon. To address this issue, we measured mood fluctuations in athletes competing in the Hardmoors 60, a 100 km, single day continuous trail-ultramarathon, and examined how variation in mood related to performance, as measured by completion time. The key finding was that the variability of athletes Total Mood Disturbance (TMD) score was significantly and positively correlated with completion time, consistent with the idea that mood is an important factor in determining race performance. Athletes also experienced a significant increase in tension immediately prior to race onset. This effect was more pronounced in less experienced athletes and significantly attenuated by measurement stage 1 at 35.4 km, which suggests the effect was driven by the release of pre-competition anxiety. Depression, anger and TMD were significantly lower at the pre-race measurement compared to the baseline measurement taken the week before. Consistent with previous studies, there were also significant increases in fatigue, anger and TMD during the race. The data are interpreted in terms of the Psychobiological model of endurance and may have broader implications for the understanding of endurance performance in other domains.

## Introduction

Trail Ultra Marathons (TUM) are footraces over a longer distance than the standard marathon distance of 26.2 miles (42.2 km). They take place over a variety of distances, time limits, terrain types and levels of support provided to participants, but all require continuous activity for periods of hours, or even days. Many factors can influence an athletes performance during a TUM, but one important factor may be their ability to regulate mood and emotional states [1–3]. Consistent with this idea, a number of studies have shown that pre-competition measures of mood, such as the Total Mood Disturbance score derived from the Profile of Mood States (POMS; [4,5]), can predict performance in short duration sports [6,7], such that better mood

**Data Availability Statement:** Data are archived on the Open Science Framework: DOI 10.17605/OSF.IO/JM3FS.

**Funding:** The research was supported by a Laidlaw Undergraduate Research Bursary awarded to PB

(The Laidlaw Foundation - Undergraduate Scholars). The funders had no role in study design, data collection and analysis, decision to publish, or preparation of the manuscript.

**Competing interests:** The authors have declared that no competing interests exist.

is associated with improved performance. However, as Beedie et al. [8] note, as the duration of the sport increases the predictive value of pre-competition measures of mood decreases because there is more scope for mood to fluctuate. In longer events such as TUMs it may therefore be more useful be to measure the *fluctuations* in mood and explore to what extent these fluctuations relate to performance. A better understanding of how mood fluctuates during TUMs may therefore give valuable insights into the psychological challenges facing athletes before, during and after TUMs.

Mood can be defined as a labile set of feelings that, unlike emotion, cannot be ascribed to a specific, known antecedent [8,9]. Empirical studies of mood states in ultrarunners have largely focused on mood immediately before and after an ultrarun, using POMS [4,5], which measures transient mood states. The scale measures six or seven dimensions of mood; tension-anxiety, depression, anger, vigour, fatigue, confusion, and esteem-related affect. These scores can be combined by taking the sum of the negative mood states and subtracting the vigour score to produce a total mood disturbance (TMD) score [10]. The most consistently reported findings are pre to post race decrease in vigour and an increase in fatigue [10–18], which is consistent with fact that completing an ultramarathon requires sustained exertion.

A significant decrease in tension has been reported in two studies [15,19]. Rauch et al. [16] attributed the decrease in tension to the achievement of their race goals on completion of the event whereas Lane, Robinson, & Cloak [19], argued that emotions such as anxiety intensify during the pre-competition period as individuals began to think about the approaching event, and the dip in tension reflects the release of this pre-competition anxiety. These two explanations make different predictions about how tension will change during a race. The 'goal completion' explanation of Rauch et al. [16] predicts that tension will gradually reduce as the athlete approaches race completion, with the largest reduction occurring at the end of the race when goal attainment is confirmed. In contrast, the 'pre-competition anxiety' explanation of Lane et al. [19], predicts that tension should decline steeply at the start of the race, as the pre-competition anxiety is released by the start of the race. Implicit in both explanations is the idea that tension at the start of the race will be elevated relative to baseline levels of tension. However, as neither of these studies took a baseline measure of tension or recorded within-race measures of tension it has been impossible to state with confidence which explanation is most likely to be correct. Tharion et al. [17] proposed a third possibility, which was that the anxiety reducing effect of exercise could be the cause of the reduction of tension, but this explanation seems unlikely, given the subsequent evidence that prolonged bouts of exercise (i.e. >180 mins of moderate intensity) do not reliably improve mood states [20]. Indeed, the extent to which pre- post comparisons fully capture the relevant psychological effects of ultrarunning on mood states remains debated [21].

Several studies [12,14,22] have attempted to bridge the gap between pre and post comparisons and within-race measurement taking daily measurements of mood during multi-stage ultramarathons. Lane and Wilson [14] measured daily changes in emotion during a 282 km six-stage ultramarathon. Participants ran stages of varying lengths on each day (16 km to 87 km) and BRUMS was used to measure emotions at the beginning and end of each day's racing. Mood states significantly changed such that fatigue increased and vigour decreased after each race. Confusion and anger also increased, but these changes were dependent on trait emotional intelligence (EI; 23), such that low scores on trait EI were associated with significant post-race increases in confusion and anger whereas high EI scores were not. Tension also declined, but only on the first and penultimate days which is consistent with a release-of-anxiety explanation for changes in tension. Broadly similar findings have been reported in other multi-day races [11,22] and other ultra-endurance events [24]. Together, these data suggest that changes in tension in multi day ultramarathons are best explained in terms of a reduction

of anxiety that occurs at the start of races. However, it is important to note that in these studies measurements were taken before and after the participants had completed the days racing, not during the activity itself. Furthermore, the changes in mood state were not related to athletes' performance. As a result, the impact of fluctuating mood states on performance during continuous TUM remains unknown.

One recent study directly addressed the issue of how mood fluctuates during a non-competitive ultrarun in 12 athletes who completed a 80.5 km treadmill ultramarathon [25]. The key observation was that those participants with high EI maintained stable mood during the race, whereas those low EI experienced a significant increase in TMD. The authors concluded that EI allows athletes to better regulate mood and pacing, although the increase in TMD was not associated with statistically significant differences in completion time. One notable difference between this study and previous work is the lack of effect on tension, which may reflect the participants' different reactions to competitive and non-competitive environments. Furthermore, in a treadmill race the terrain and environmental conditions are stable, whereas in a real race these factors may vary considerably and this uncertainty might also affect mood. It therefore remains unknown to what extent the results of non-competitive, treadmill running will generalise to competitive races.

To summarize, research has largely focused comparisons of mood before and after long races. There is clear evidence that mood fluctuates during races, but few studies have explored how these changes in mood are related to performance. To address these issues, we examined mood fluctuations in athletes competing in a 103 km TUM. Mood was assessed one week prior to competition, on the day of competition prior to race onset, at each stage within the competition and after race completion. A key question was whether or not variation in mood would be associated with changes in performance. Based on previous findings, we also hypothesized that there would be significant changes in mood states of tension, vigour and fatigue during the trail-ultramarathon, such that (a) tension would increase between the week-before and pre-race measurement, reflecting the onset of pre-race anxiety, then decline, with the steepest decline occurring between pre-race measurement and Stage 1, (b) that vigour would peak pre-race then decline, and (c) that changes in fatigue would be smallest during the middle part of the race.

## Methods

### Race details

For consistency of nomenclature, the race title is based on the distance ran in miles, however, this paper will adopt the use of Kilometres (km) throughout this paper in line with the position statement from the Ultra Sports Science Foundation [26]. The Hardmoors 60 (HM60) race is permitted under the rules of the Association of Running Clubs [27], which includes trail running. The HM60 race consists of a single-day 100 km semi-supported continuous trail-ultramarathon which is a part of the Cleveland Way national trail in England, predominately following coastal tracks with specifically; 87% on paths, 3% on tracks and 10% on roads. The race has a starting altitude of 99 m and finishing at 37 m above sea level, with a with a total positive and negative elevation of 2300 m and 2350 m, respectively. Participants were required to complete the race within 18 hours and there were cut-off times for each checkpoint [28].

### Participants

Thirty-four participants (19 males and 15 females, $M$ age = 44, $SD$ = 8) volunteered to take part in the study after replying to adverts on the race groups social media page across two versions of the HM60 (September 2018; September 2019). Demographic information is presented

Table 1. Demographic information about race participants.

| | n (out of N = 30) | Mean | SD | Range | Min | Max |
|---|---|---|---|---|---|---|
| Longest race completed (km) | 30 | 152 | 73.71 | 187 | 21 | 257 |
| Previous Ultra's over 80km | 25 | 10 | 9.78 | 39 | 1 | 40 |
| Previous Ultra's over 160km | 15 | 1 | 1.87 | 5 | 1 | 6 |
| Reported previous DNF | 20 | 2 | 1.66 | 4 | 1 | 5 |
| Training Km PW** | 30 | 49 | 11.81 | 91.9 | 12.1 | 104 |
| Race position* | 30 | 118 | 44.03 | 180 | 26 | 206 |
| Finish time (mins) | 30 | 966 | 83 | 327 | 781 | 1108 |

[a]DNF: Did Not Finish.

[b] Average number of kilometers run per week (over 12 weeks preceding the race).

[c]Out of 206 total finishers.

in Table 1. All participants gave written informed consent. The study was approved by the Durham University Dept of Psychology Research Ethics Committee. Two participants did not complete the race, four did not complete the Week Before measurement (but did complete all the race-day measurements) and two did not complete all the questionnaire items at all the stages.

## Materials

The 24-item Brunel Mood Scale (BRUMS; 29) with the stem 'How do you feel right now' was used to measure mood states. The BRUMS is a shortened version of the POMS developed to facilitate the collection of mood data before and during competition [28]. The scale uses 24 single adjectives which produce six subscales: anger, confusion, depression, fatigue, tension, and vigour. The scale is scored using a 5-point scale (0 = not at all, 1 = a little, 2 = moderately, 3 = quite a bit, 4 = extremely). Total Mood Disturbance (TMD) was calculated as the sum of the negative mood scores (anger, confusion, depression, fatigue, tension) minus vigour.

## Procedure

Participants were sent electronic versions of the BRUMS and demographic survey to be completed in the week before the race. On race day, the participants were met by the research team during registration for the event and asked to complete the pre-start BRUMS survey. The research team verbally stating the phrases and then manually recording responses on a paper copy of the questionnaire. The pre-race measure was taken between 30 and 90 minutes prior to the race starting. During the race participants were met by a member of the research team just as they entered checkpoint 3 at 35.4 km and checkpoint 6 at 66 km at which points the BRUMS was verbally administered. The final BRUMS measurement was verbally administered as finishers entered the finish hall. This occurred no longer than 15 minutes after race completion.

## Statistical analysis

Data were analyzed using JASP version 0.9.1 (JASP Team 2021). Alpha level was .05 for inferential statistics.

For the analysis of mood changes during the race data from the 26 participants for whom we had full data-sets were analysed with a 5 (Measurement Point, 1 Week Before (1WB), 0 km, 35.4 km, 66 km, 100 km (Finish) x 6 (Mood Factor: anger, confusion, depression, fatigue,

tension, vigour) repeated measures ANOVA. Statistically significant interactions were explored using repeated measures ANOVAs with a single factor of stage for each mood state. Total mood disturbance was subjected to a One-Way ANOVA with a single factor of stage. Significant main effects were followed up with paired samples t-tests comparing adjacent measurement points. Alpha levels were adjusted to control for multiple comparisons using a Bonferroni correction.

Two analyses were conducted to examine the relationship between mood and performance. Firstly, following the majority of previously published studies [7] we looked for an association between pre-competition TMD and the time taken to complete the race using a Pearson correlation. Secondly, we calculated the standard deviation of each individual's TMD scores from all measurement points to obtain an estimate of mood variation during the race ($TMD_{SD}$). In the four cases where no 'Week Before' measure was obtained we used the SD of the remaining measurement points. Inspection of the boxplots indicated that participant 16 was an outlier and this participant was excluded from further analysis.

An exploratory analysis examined whether experience in running ultramarathons was related to mood changes during the run. For this analysis we included data from the 4 participants who did not complete the 'week before' measurement and excluded data from this measurement point resulting in sample of 30. We used a median split (Md = 5.5 races) to create a group of less experienced runners (n = 15, mean age 41, mean weekly mileage 50 km, mean number of completed 80 km races 1.8, SD = 1.7, Range 0–5) and a group of more experienced runners (n = 15, mean age 47, mean weekly mileage distance 49 km, mean number of completed 80 km races 14.5, SD = 10.4, range = 7–40). We then conducted an exploratory mixed 6 x 2 ANOVA with within-subjects of Measurement Point and between-participants factor of Experience for each mood factor. The main effects of Mood Factor have been described in the preceding analysis so are not reported again.

## Results

### Mood changes

There was no significant main effect of Measurement Point; $f_{(4,100)} = .49$, $p = .74$, $\eta^2 p = .02$. There was a large and significant main effect of Mood; $f_{(5,125)} = 76.61$, $p < .001$, $\eta^2 p = .75$, and a significant Measurement Point x Mood interaction; $f_{(20, 500)} = 23.11$, $p < .001$, $\eta^2 p = .48$. These effects and interactions are illustrated in Fig 1.

There were small but significant main effects of stage on anger ($F_{(4,100)} = 5.82$, p = < 0.05, $\eta^2_p = .09$) and depression ($F_{(4,100)} = 3.94$, p = < 0.05, $\eta^2_p = .09$), and large effects of stage on fatigue ($F_{(4,100)} = 28.25$, p = < 0.01, $\eta^2_p = .53$), vigour ($F_{(4,100)} = 22.35$ p = < 0.01, $\eta^2_p = .47$), and tension ($F_{(4,100)} = 11.76$, p = < 0.01, $\eta^2_p = .32$).

Anger significantly decreased between Week Before and 0 km ($t_{(25)} = 3.01$, p < .01, d = .59) but none of the other contrasts were statistically significant. Depression also significantly decreased between Week Before and 0 km ($t_{(25)} = 3.54$, p < .01, d = .69) then increased. The increase between 0 km and 35.4 km was not significant following Bonferroni correction ($t_{(25)} = 2.56$, p = .017, d = .5) and none of the other comparisons were statistically significant. Fatigue decreased between week before and 0 km, but the effect was not statistically significant when the Bonferroni correction was applied ($t_{(25)} = 2.24$, p = .034, d = .44). Fatigue significantly increased between 0 km and 34 miles ($t_{(25)} = 4.00$, p < .01, d = .79) and between 66 km and 100 km ($t_{(25)} = 5.33$, p < .01, d = 1). However, there was no statistically significant change in fatigue between 34 and 66 km ($t_{(25)} = 1.79$, p = .09, d = .35). Vigour significantly increased between Week Before and 0 km ($t_{(25)} = 2.97$, p < .01, d = .58) then decreased between 0 and 35.4 km ($t_{(25)} = 4.28$, p < .01, d = .8) and between 66 km and Finish ($t_{(25)} = 5.19$, p < .01,

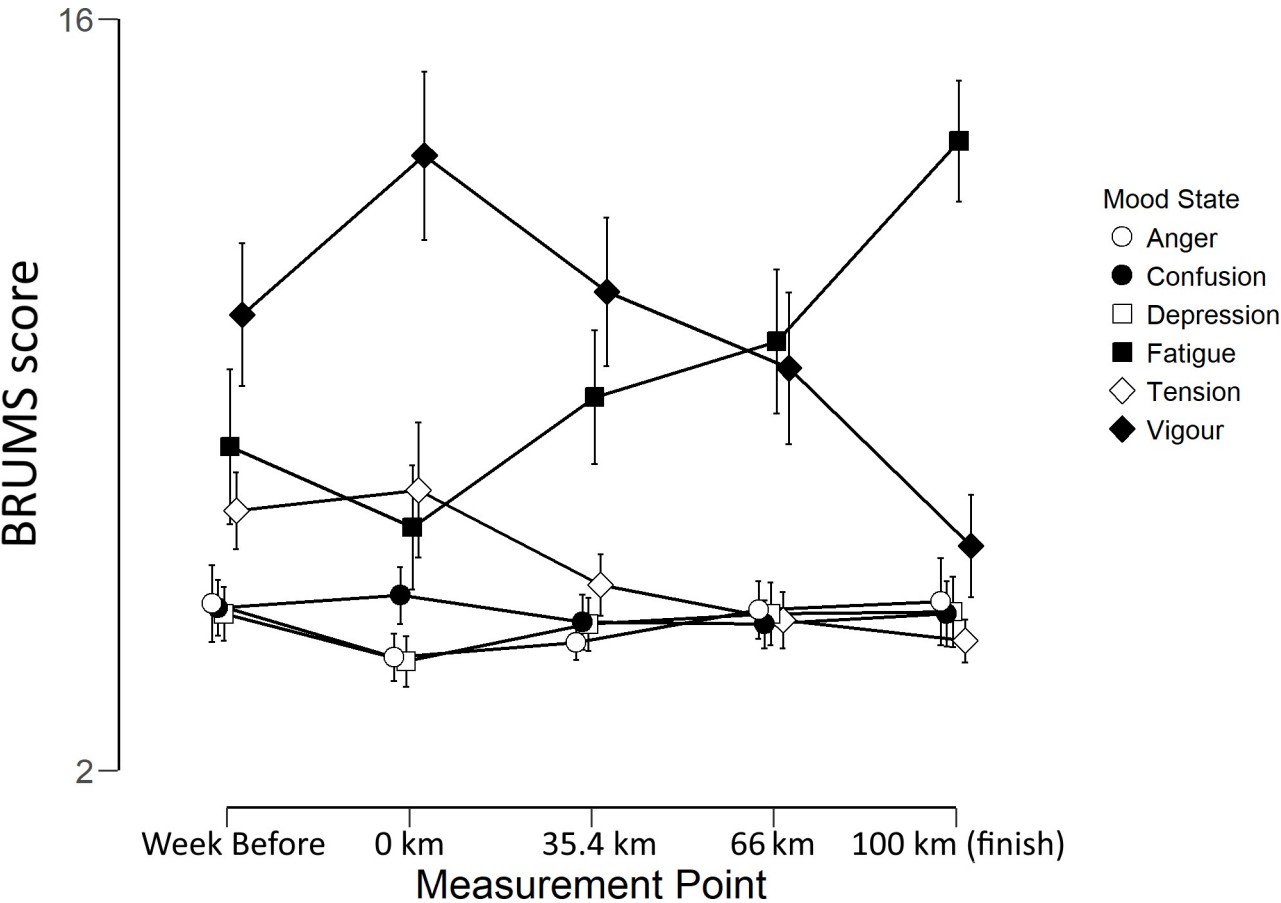

**Fig 1. Mean BRUMS scores for each Mood Factor at each Measurement Point (km).** Error bars show 95% confidence intervals.

$d = 1$). However, as with fatigue, there was no statistically significant change between 34 and 66 km ($t_{(25)} = 1.9$, p = .07, $d = .37$). Tension significantly decreased between 0 km and 35.4 km ($t_{(25)} = 3.15$, p < .01, $d = .62$). None of the other comparisons were statistically significant after the Bonferroni correction was applied.

The analysis revealed a significant main effect of Measurement Point ($f_{(4, 100)} = 15.7$, p < .01, $\eta^2_p = .39$). Bonferroni corrected post-hoc t-tests revealed significant reduction in TMD between Week Before and 0 km ($t_{(25)} = 3.31$, $p < .01$, $d = .65$) and a significant increase in TMD between 66 km and Finish ($t_{(25)} = 5.09$, $p < .01$, $d = .98$). These effects are illustrated in Fig 2.

## Mood fluctuation and performance

The association between the two pre-competition TMD and performance was not statistically significant ($r_{(28)} = .02$ $p = .93$). However, the variability of Total Mood Disturbance during the race (TMD_{SD} scores) was significantly correlated with the total time taken to run the race ($r_{(28)} = .42$, p = .025), consistent with the idea that the ability to manage mood fluctuation is associated with better race performance (see Fig 3).

## Effects of experience

There was no main effect of Experience for any of the factors. The only statistically significant interaction between measurement point and Experience was observed for tension ($F_{(3, 84)} =$

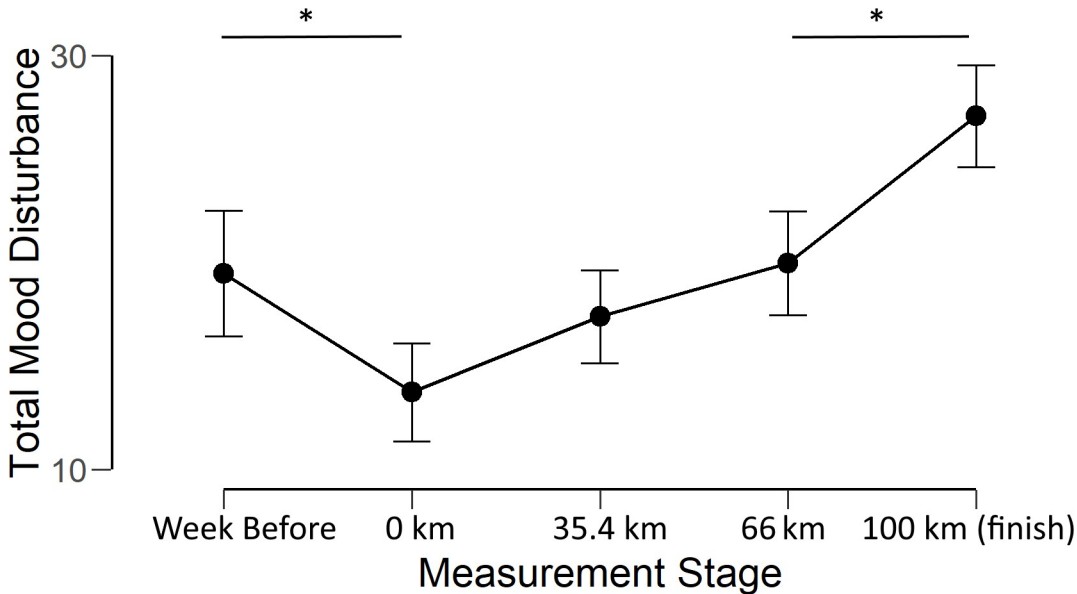

**Fig 2. Total Mood Disturbance score at each measurement stage.** Error bars show 95% confidence intervals. * indicates a statistically significant difference.

6.18, p < .001, $\eta^2_p$ = .37). Inspection of Fig 4 suggests that this effect is due to the less experienced group experiencing higher tension than the more experienced group at measurement point 0 ($t_{(28)}$ = 2.71, $p$ = .011, $d$ = .99).

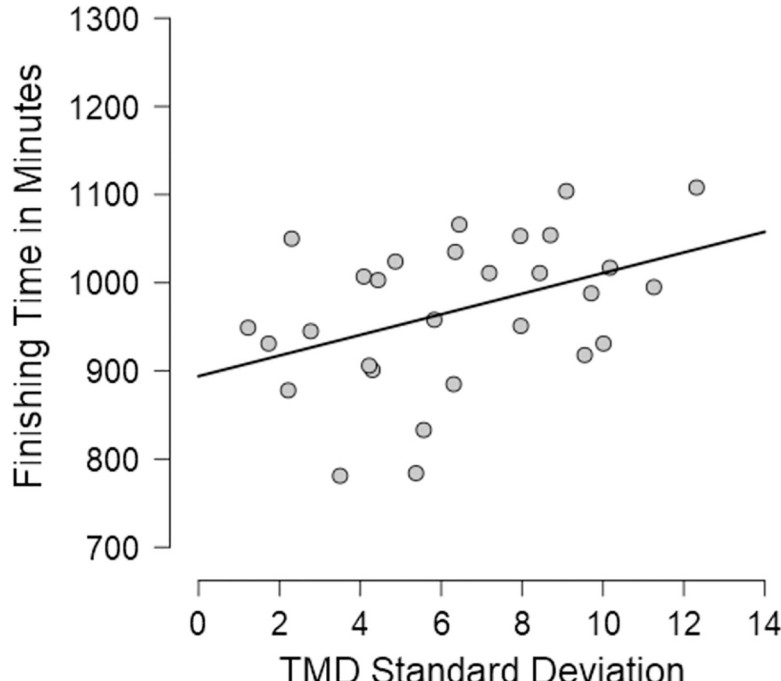

**Fig 3. Scatterplot showing the significant, positive correlation between TMD standard deviation scores and finishing time.**

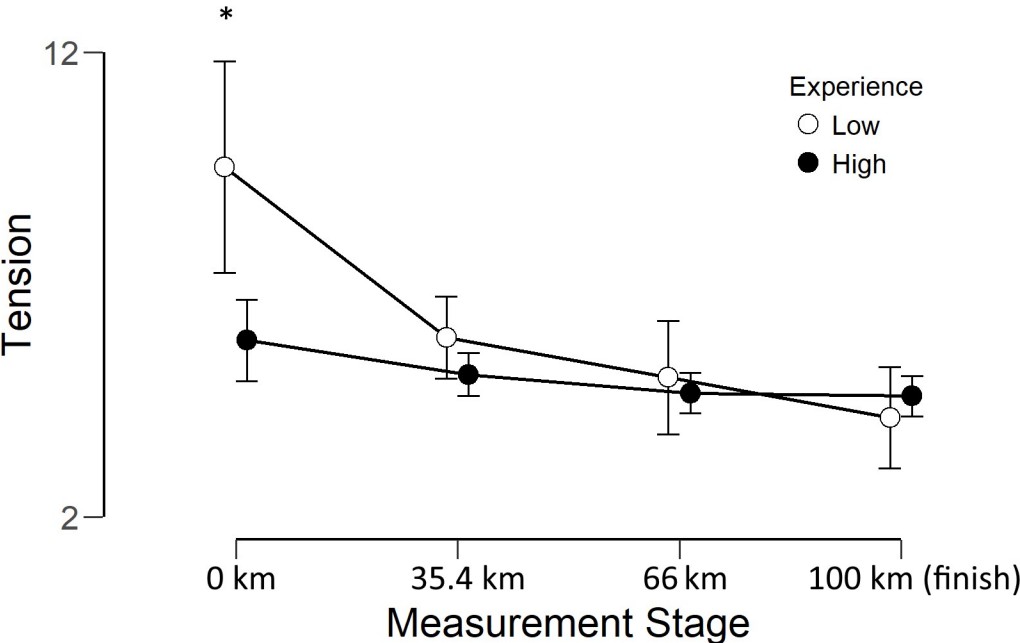

**Fig 4. Tension scores for less experienced (○) and more experienced (●) athletes at each measurement point.** Error bars show 95% confidence intervals. * indicates a statistically significant difference.

## Discussion

This study was designed to test the idea that competing in a trail-ultramarathon would be associated with significant changes in mood, and that these changes would be related to athletes' performance. They key finding was that the variability of TMD scores was significantly and positively associated with the time taken to complete the runs. It was also observed that tension significantly declined between pre-race and 35.4 km, but not between the other stages, that Vigour peaked at the pre-race measurement then declined at subsequent measurement points, and that anger and depression were significantly lower at the pre-race measurement compared to week before. Consistent with these patterns, TMD had a v-shaped profile, such that mood was best at the pre-race stage.

An unresolved question in the literature was the extent to which changes in mood play a functional role in athletic performance in ultrarunning. The analysis of the variability of TMD scores clearly showed that increased TMD was associated with poorer performance (as indexed by time taken to complete the race). This is, we believe, the first empirical demonstration that increased variability in mood is related to impaired performance during ultrarunning. One possible explanation for this effect is that greater fluctuations in mood requires more effortful self-regulation which results in increased mental fatigue. Mental fatigue increases the perception of effort, which has consistently been associated with reduced endurance performance [30,31]. Furthermore, according to the Psychobiological model [32], when the perception of effort is high, individuals' must make an evaluation about whether they believe that the goal is still achievable. If they decide their goal is unachievable they either re-evaluate their goal, for example to reduce the pace, or to quit. In this view, the psychological mechanism by which increased mood variability impacts performance is the lowering of individual goals about performance caused by an increased perception of effort. An appealing feature of this account is that it is amenable to future empirical investigation, as it clearly predicts

that athletes with high mood fluctuations should report increased perception of effort, increased mental fatigue and more frequent revisions to their performance goals.

The finding that tension showed a steep decrease between 0 and 35.4 km, rather than a gradual decrease throughout the race is hard to reconcile with the suggestion that changes in tension are due to goal attainment [17] which, because goal attainment cannot occur until the race is completed, would predict that the biggest change in tension should occur at the end of the race. Furthermore, the profile of tension scores for less experienced runners showed a distinct peak at the pre-race measurement which was not seen in the more experienced runners (Fig 2). Increased experience is associated with reduced pre-competition anxiety [33,34], which might account for the reduced pre-race tension in this group. It might be argued that this explanation is unlikely because ultra-runners tend to be motivated by personal goal achievement rather than competition [35], so are unlikely to be affected by pre-competition anxiety. However, while ultrarunners are primarily motivated by goal achievement, they report similar levels of competitive motivation as non-ultrarunners, so it is not the case that competitive goals are irrelevant to these athletes *per se*. Interestingly Waśkiewicz, et al. [35] report that the relationship between experience and competitive motivation is negative, so it is plausible that the less experienced athletes were more likely to be motivated by competition and therefore more likely to experience pre-competition anxiety. Overall, the data seems consistent with the idea that changes in tension observed in previous studies reflect the release of pre-competition anxiety [19] rather than goal attainment. Indeed, the study which is most directly comparable to ours [25] reported no change in tension during a timed but non-competitive 80 km treadmill run, perhaps because negative emotions such as anxiety are typically stronger before competitive races compared to non-competitive events.

Vigour peaked at the pre-race measure then significantly decreased, which was similar to the pattern observed by Howe et al. [25]. This effect likely arises due to a combination of the feelings of excitement and physiological arousal athletes experience before the race begins [36], combined with the benefits of tapering [37]. Tapering involves significantly reducing training load in the week (or weeks) prior to a race. As vigour measures perceptions of how active and energetic the person feels it follows that reducing the volume of physically demanding activity could increase alertness and energy level and therefore vigour.

As with Howe et al. [25], we observed changes to depression and anger, such that these scores dipped to their lowest level just before the race began. In our case the decrease in anger and depression was statistically significant compared to the week-before measure, reflecting the increased power of our larger sample (N = 12 and N = 26 respectively). The TMD scores were also significantly lower at 0 miles compared to the week before. These findings demonstrate that TUM runners experienced significantly enhanced mood on the day of a race, relative to the week before. As noted earlier, some of this enhanced mood can be attributed to enhanced sense of energy due to tapering and excitement about the race beginning, but the significant reductions in anger and depression indicate that both reduction in negative states and improvement in positive states contribute to this effect. Contrary to Howe et al. [25] we did not observe a statistically significant increase in depression or confusion during the race. The reason for this discrepancy is not clear, but differences in context (our participants completed a competitive, outdoor ultrarun whereas those of Howe et al. [25] completed a non-competitive, indoor time-trial) account for the differences. For example, it may be that some of more negative emotions such as depression and confusion are mitigated by factors such as the presence of other athletes, the presence of support teams, or some other factor specific to different types of ultra-endurance event. Indeed, the role of these factors on mood are not well understood and may be a fruitful avenue for future research. Furthermore, this discrepancy

emphasises the importance of defining ultramarathons according to the criteria proposed by Scheer et al. [26].

There are some limitations to the study. Firstly, the sample was collected over two races. Although the route and terrain were similar, other environmental factors such as the weather were different and this may have led to differences in mood states on the day of the race. Secondly, the window for data collection at the pre-race measurement was somewhat large at 90 minutes, so it is possible that some participants were tested outside the optimal window for predicting performance. Thirdly, the mode of data collection meant that the BRUMS data was not anonymous, which may have introduced subtle observer effects. This could be an issue if, for example, some participants under-reported negative mood states such as fatigue, which might have resulted in an underestimation of the variability of negative mood factors.

## Conclusions

The key new finding was that variation in TMD was significantly associated with performance, such that increased TMD variability was associated with slower running. This negative relationship between TMD and performance was interpreted in terms of the Psychobiological model of performance [32], such that the higher cost of self-regulation due to mood fluctuation produces mental fatigue which leads to increased perception of effort, resulting in a reduction in pace. A second important finding was that the changes in tension which occurred at the start of the race were most prominent in less experienced athletes, probably because less experienced ultrarunners tend to hold more performance-oriented goals and experienced greater pre-competition anxiety. These findings could have important implications for researchers studying and developing strategies aimed at improving athletes' ability to manage their emotional states in order to optimise performance during TUMs and in endurance sport more broadly [e.g., 38–40].

## Supporting information

**S1 File. Supplementary analysis of within race data.**
(DOCX)

## Acknowledgments

Thanks to Ann Brown, Jane Raper, Naomi Roopchand and Lee Burgum for support with data collection. We would also like to thank Dr Jennifer Sherwood for her thoughtful and constructive comments on a previous version of this manuscript.

## Author Contributions

**Conceptualization:** Paul Burgum, Daniel T. Smith.

**Data curation:** Daniel T. Smith.

**Formal analysis:** Paul Burgum, Daniel T. Smith.

**Funding acquisition:** Paul Burgum.

**Investigation:** Paul Burgum.

**Methodology:** Paul Burgum.

**Project administration:** Daniel T. Smith.

**Supervision:** Daniel T. Smith.

**Writing – original draft:** Paul Burgum, Daniel T. Smith.

**Writing – review & editing:** Paul Burgum, Daniel T. Smith.

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
