## [Decision Letter · Decision Letter 0]

14 Apr 2021

PONE-D-21-06919

Mood fluctuations during a 103 km ultramarathon

PLOS ONE

Dear Dr. Smith,

Thank you for submitting your manuscript to PLOS ONE. After careful consideration, we feel that it has merit but does not fully meet PLOS ONE’s publication criteria as it currently stands. Therefore, we invite you to submit a revised version of the manuscript that addresses the points raised during the review process, mainly the ones from reviewers 2 and 3.

We look forward to receiving your revised manuscript.

Kind regards,

Pedro Tauler, Ph.D.

Academic Editor

PLOS ONE

Journal Requirements:

Reviewers' comments:

Reviewer's Responses to Questions

**Comments to the Author**

1. Is the manuscript technically sound, and do the data support the conclusions?

Reviewer #1: Yes

Reviewer #2: Yes

Reviewer #3: Partly

Reviewer #4: Partly

2. Has the statistical analysis been performed appropriately and rigorously? 

Reviewer #1: Yes

Reviewer #2: Yes

Reviewer #3: No

Reviewer #4: No

3. Have the authors made all data underlying the findings in their manuscript fully available?

Reviewer #1: Yes

Reviewer #2: Yes

Reviewer #3: Yes

Reviewer #4: No

4. Is the manuscript presented in an intelligible fashion and written in standard English?

Reviewer #1: Yes

Reviewer #2: Yes

Reviewer #3: No

Reviewer #4: Yes

5. Review Comments to the Author

Reviewer #1: An interesting study that examines mood state changes in ultra-endurance. Using the POMS model of mood, a series of hypotheses on how moods states will change are presented and tested. The author should focus on competitive mood throughout the article and so discussion of mood in exercise is out of scope of the study.

The authors report mood state changes in treadmill studies. Whilst the distance is known, so is the terrain whereas in a real race, the terrain and environment conditions contribute to uncertainty. This connects to the notion that the authors should relate their findings to competitive settings. I believe these revisions are easily overcome.

The authors might wish to consider how they use the term mood, which are feelings which do not have a specific antecedent that is known to the participant, from emotion, where the cause of feelings is known (Beedie et al., 2010). It is possible for mood and emotion to change during an ultra.

The researchers delimit the work to running. This is understandable. Yet, they have non-competitive running, which is also understandable. However, it made me wonder why they did not use research from ultra events that were not running, such as cycling. Is there a huge difference in mood states changes between ultra-endurance events? I am not sure, but given the exploratory aspect of this study, such proposals could be considered.

Lane and Terry (2000) focused on depressed mood and argued that a no-depression – depression dichotomy was influential. Your data would allow the possibility to test whether an increase in depression, results in a much larger increase in negative mood. Whilst running will lead to an increase in fatigue, could depression accelerate perceptions of negative mood.

Relevant references to consider

1. Beedie, C. J., Terry, P. C., & Lane, A. M. (2005). Distinguishing mood from emotion. Cognition and Emotion, 19, 847-878.

2. Lahart, I., Lane, A.M., Hulton, A., Williams, K., Charlesworth, S., Godfrey, R., Pedlar, C., George, K., Wilson, M., & Whyte, G., (2013). Challenges in maintaining emotion regulation in a sleep and energy deprived state induced by the 4800km ultra-endurance bicycle race; the Race Across America (RAAM). Journal of Sports Science and Medicine, 12, 481-488. http://www.jssm.org/vol12/n3/17/v12n3-17text.php

3. Lane, A. M., & Terry, P. C. (2000). The nature of mood: Development of a conceptual model with a focus on depression. Journal of Applied Sport Psychology, 12, 16-33.

4. Lane, A. M., Whyte, G. P., Shave, R., Barney, S., Stevens, M. J., & Wilson, M. (2005). Mood disturbance during cycling performance at extreme conditions. Journal of Sports Science and Medicine, 4, 52-57. http://www.jssm.org/vol4/n1/7/v4n1-7text.php

5. Pedlar, C. R., Lane, A. M., Lloyd, J. C., Dawson, J., Emegbo, S., Stanley, N., & Whyte, G. P. (2007). Sleep profiles and mood state changes during an expedition to the South Pole: a case study of a female explorer. Wilderness and Environmental Medicine, 18, 2, 127-132.

6. Stanley, D. M, Lane, A. M., Beedie, C. J., Devonport, T. J. (2012). “I run to feel better; so why I am thinking so negatively”. International Journal of Psychology and Behavioral Science, 2, 6, 28-213. 10.5923/j.ijpbs.20120206.03

Reviewer #2: This study examined fluctuations in emotions during a TUM. The manuscript addresses a gap in the literature, which only compared pre to post UM changes in mood. The authors found decreases in vigour and tension and increases in fatigue during the race, while there were no changes in anger, depression or confusion. A comparison between low and high experienced subjects revealed that the increase in tension on the day of the race was significantly higher in the low experienced group compared to the high experienced group. Same applies for the decrease in tension between

race onset and 34 km.

Introduction:

• “the literature exploring the consequences of different types, durations and intensities of exercise on mood after bouts of exercise has reached a broad consensus that the optimal level of exercise for mood benefits in healthy adults is 20-30 minutes of moderate intensity exercise” -> Careful with this statement. There are also studies that show that higher intensities led to the highest increases of positive mood (Schmitt et al. 2020, Bartlett et al., 2011). It is not yet clear what role the fitness of the subjects plays when it comes to mood. In addition, many studies have often only examined moderate intensity instead of comparing different intensities.

• double statement: “The study of mood states in ultrarunners has largely focused on mood immediately before and after an ultrarun, using the Profile of Mood States” and “Numerous studies have measured these dimensions before and after an ultramarathon and reported consistent findings.”

• The introduction is very long at should be shortened by half.

Methods:

• Table 1: Abbreviations not explained (DNF, PW).

• “All participants gave informed consent, and the study was approved by the Durham University Psychology Research Ethics Committee.” Already mentioned in “Participants” paragraph

• “Data were collected on the week prior to and during the 2018 and 2019 Hardmoors 60 races.” -> on the week prior and during is doubled in the following paragraph. Leave it out in this sentence -> Data were collected during the 2018 and 2019 Hardmoors 60 races.

• The acquisition of the pre-race measurement “30 minutes and 1.5 hours prior to the race starting” gives potentially high variance as e.g. tension might be higher 30 min before race that 90 min before race. This large timeslot/lack of standardization should mentioned in the limitations.

• You should think of performing a subgroup analysis with the five that did not complete the BRUMS for the Week Before measurement (but did complete all the race-day measurements) as one group without median split as additional information. Could be implemented as supplement.

Results:

• Why are only 30 subjects in the median split analysis? 26+5 should result in 31 subjects.

Minor:

• “The study was approved by the Durham University Dept of Psychology Research Ethics Committee” -> period at the end of the sentence is missing

• “which produce six subscales that:” -> delete “that”

• “Eight participants were removed from the analysis.” -> replace . by :

• “(Mood Factor: anger, confusion, depression, fatigue, tension, vigour). Repeated Measures ANOVA.” ->remove period after bracket: …vigour) Repeated Measures…

General comments:

The manuscript contains unnecessary duplications and should be revised in this respect.

Reviewer #3: This article entitled “Mood fluctuations during a 103km ultramarathon” aimed to determine the impact of a single-day event continuous trail-ultramarathon on mood fluctuation. The key findings support that of previous studies in identifying pre-event vigour and reduced anger, depression and fatigue and as participants competed in the event, there was a decline in vigour and an increase in fatigue as expected. There were also increases in tension at the start of the event that declined as the event progressed.

However, it was not really clear the implications and association these changes in mood would have on performance success or participant adherence or motivations for example.

The work would have been strengthened with some comparative analysis to establish the impact of such patterns on either performance times, pacing, completion, drop-out/adherence, etc. The reader is left thinking, ‘so what’? Overall, the authors should consider the impact of why during race event mood fluctuations have importance?

This manuscript could be significantly improved with reworking the data and interpretation of the results with more robust analysis and interpretation in the discussion to consider the application of the findings.

The methodology is clear however improvements to details can be made and include more clarification on the data analysis and statistical approaches rather than in the results (see reviewer suggestions).

General data analysis of Mood scores and statistical reporting is accurate but there are mistakes in the data type reporting (ordinal) and calculation of performance times in table 1. It is unclear to the reviewer why the authors used sub-analysis within their data to compare ‘low experience’ athletes and ‘high experience’ athletes and they do not clarify this analysis and methodological decision in the methods section.

Data are deposited in the Open Science Framework and a pre-print version cited in sportrxiv. The authors would benefit from consider the scientific rigour and depth of the pre-print version to better reflect and improve the quality of the current submitted manuscript.

Generally the manuscript is appropriately formatted and presented however there are a number of grammatical, structural and presentation issues that can be reviewed to improve the clarity and accuracy of the work.

The authors cite and are aware of the relevant and seminal works of the topic area and critique and incorporate key studies linked to the work, however the reference list is not according got the Vancouver style and contains many errors and unused citations.

The authors should reconsider a robust reworking of the manuscript for accuracy and the analysis and interpretation of their findings with more application and are then encouraged to reconsider submitting a revised version with a more detailed critique and depth of the discussion particularly.

Amendment suggestions:

Abstract

Line 15-16- the introductory line is unimpactful and not really relevant or necessary to the overall aim and focus of the work. Physical activity is to vague and sweeping to the relevance of ultra-endurance events. Consider rewording or expanding the implication towards ultra-endurance and why mood is of importance particularly during an event.

Line 23/ 31- It is not necessary or adding to the abstract to use the TUM abbreviation. I would remove.

Indicate the event is a single-day race.

The terminology or classification of ‘low experience’ is uncommon, perhaps something like novice or inexperienced ultra-runner would be clearer. Also ‘highly’ experienced is again presumptive, so perhaps ‘more’ experienced.

Reanalysis of the data will enable the summary to be more indicative of the implications and effect mood fluctuations DURING an event may play importance to runner success /performance for example.

Line 34- clarify what you mean by ‘in other domains’.

Introduction

Line 41- replace citation ‘Arent et al., 2000’ with a more relevant citation to the study focus.

Line 46- you indicate and relate the importance of understanding of mood during long duration bouts and rationalise due to increases participation in ultra-endurance events but expand to clarify why this may be of importance. For example, is it likely to impact performance success etc.

Not sure your abbreviations TUM and TUR are necessary.

Line 50- update this data and citation with the more recent dataset that is available.

Line 53 – add supportive citation(s).

Line 58- ‘The study’ is grammatically incorrect, check sentence structure.

Lines 58-62. In addition to the citations that support the tools of assessing mood, the sentence should also be supported with citations of at least example studies where these tools have been specifically used in ultra-runner studies. You could consider removing unnecessary citations and only citing the seminal citations (.e.g McNair et al., 1971; 1992) for the scale development. Also indicate the scale (POMS) is used to assess transient mood states. Some of these citations could be relocated to your methods section instead.

Line 64- Clarify that TMD is a global indicator and in methods indicate how this is calculated.

Lines 65-66 is vague and unsupported in depth.

Line 65- ‘Numerous studies have…’ should be supported by citations, and identify the seminal or key examples as more than one should support this claim ‘numerous’. Line 66- ‘reported consistent findings’, you need to expand this claim/statement with what exactly they report or identify as key outcomes.

Line 66-68 is supported with an irrelevant citation and the sentence claims inaccurate reflections of ultra-marathon requirements. Replace ‘Baron et al., 2009’ with a more suitable citation.

Line 68 – ‘perception of physiological resources’ needs better clarification and detail around what this means, particularly in relation to ultra-endurance performance.

Line 75-80 – the grammatical structure and clarity of the sentences needs improving. It is unclear what is being stated. Furthermore, this needs more clarification/explanation to why in-event monitoring would be beneficial to Weiner’s position and also how it relates to ultra-marathon performance/success/outcomes.

Line 81-83 - Vague sentence start. Instead of claiming ‘others’ cite who those ‘others’ are. Furthermore the sentence is unclear and needs rewording and checking the grammatical structure.

Line 83-85- do athletes ‘begin’ or have available to them to implement strategies and what sort of strategies are used. Expand clarity and detail here.

Line 85- Hanin, 2010 citation missing from reference list.

Line 85- ‘…few studies’ support or indicate what studies (citations) do exist to support this statement.

Line 85- again support / expand to explain why a baseline compared to pre-race measure of emotion would be beneficial.

Line 86-89 – expand clarity to why a within event indication of ‘tension’ would be useful. Elaborate on the Lane et al., explanation (and include citation date), and ‘Goal completion account’ needs a supportive citation and elaboration.

Line 89- ‘…or follows some other profile’. What is meant by this, expand and give examples of such possible profiling.

Line 90- ‘anxiolytic effect’. Also expand what is meant by this term.

Line 92- ‘….produce negative’, check grammatical sentence structure.

Also be critical as ‘Reed and Ones, 2006 is limited in interpretation and ultra—distance events are very different to acute shorter duration bouts.

Overall from this point, the introduction needs more depth of the importance of all this detail and the emotion and profiling to ultra-marathon running and performance, such as success.

Line 96- ‘Several studies’ again this claim needs citation support.

Line 100- Clarify the abbreviation ‘BRUMS’ and include the seminal citation (Terry et al., 1999; 2003) for this short version scale, and also explain it is a short version of POMS etc. Move the detail cited on page 7 lines 127-128 to here.

Line 101- ‘Consistent with previous work…’ support this claim with suitable citations.

Lines 102-104- ‘Confusion and anger changed’ was quite vague and unclear what sort of change and why this was of importance. Also include citation for TEI (Petrides, 2001). Expand briefly what TEI represents/indicates.

Line 105- ‘1st’ – write as ‘first’.

Line 105-106 is unclear. What is meant when you say ‘consistent with a release-of-anxiety explanation…’ Check clarity and amend this sentence.

Line 106-Punctuation- ‘..’ at end of sentence.

Line 108- what is ‘non’? Better terminology use needed here.

Line 108- spelling mistake ‘sub-arctic’.

Line 109- degrees ‘Celsius’.

Considering the detail, for example at Lines 110-onwards. To help your result explanation and interpretation perhaps needs to relate to Morgan’s Mental Health Model (1980;1985) iceberg-type profile, to help better clarity in explaining the effect/pattern of data seen.

Line 111- spelling ‘in in’

Line 112- grammar of sentence ‘in similar’.

Line 115- Rausch Goal Attainment needs citation.

Line 116-‘ anxiety account’ is unclear and needs clarification of what is meant by this sentence detail.

Line 116-117- ‘similarly harsh’ – appreciate the implication here, but also be critical of the fact its one temperature extreme to another.

Line 117- ‘to the other multi-day studies’. Remove ‘other’ and also support with example citations.

Lines 116-119- short sentence use. Section can be more concisely written.

Line 121- ‘the impact of physical activity’ I think better to term ‘physical exertion’ is better refection of the efforts of ultra-endurance events.

From this point, again there needs expansion to why during race event mood states are of importance or what sort of impact/relevance it is likely to have.

Line 140- grammar ‘will’ to ‘would’.

Line 141-143- Citation needed to support sentence.

Line 146- ‘decrease’ not ‘decreases’.

Line 148- ‘a significant improvement’.

Line 154- ‘first’ not ‘1st’.

Your critique on lines 129-157 – consider the impact of the data.

Line 160 ‘ measurements’.

Lines 161-162 contradicts the subsequent lines 163-164. So make it clear where the differences or lack of evidence lies and clarify what are the novel aspects to your research and why this is of importance/relevance.

Methods

Race details

Indicate the event is a single-day race with an 18 hour time restriction to complete the distance with check-point cut-off times.

Line 179- define abbreviation (HM60)

Line 180- ‘The Association of Running Clubs’ (2021). Check grammar or amend citation list detail

Lines 181-183- remove this detail as does not add to the focus and interpretation of the work.

Lines 183- 185- include citation/source for detail.

Participants

The detail provided in results (lines 224-226 etc.) is better reported in the methods. Again proofread to remove short sentence use and be more concise.

Line 195- full stop missing.

Lines 191-192- Considering two races across two years were used to recruit participants, it is importance to acknowledge the race conditions for these two years (and whether or Not race checkpoints/regulations were the same) as the weather for example may impact the mood states during the events and therefore is a potential limitation to the grouping of your data and mood fluctuations. Also acknowledge and identify this critique in your discussion.

Table 1 (page 11).

Need to footnote and define what DNF abbreviation is.

Include units of measure for each variable.

Also report n= numbers for example, did all participants in the data set complete ultras over 80km or 160km- it may be more informative to report of the cohort the n= or % (frequency).

Again for DNFs’ it may be better to report of cohort, how many reported DNFs and n=events (frequency).

Regarding race position, did all participants complete /finish and place.

Your race position ordinal numbers cited in the table need to be rounded to whole numbers – you cannot finish in 118.19th place

Check and reanalysis your finish time data (reported as mins) as this is not accurate. 96 mins to complete 103kms!?

Materials

More depth/detail of the scale needed. Indicate the short-version form of the POMS was used. Clarify the six scales calculated and the TMD (Heuchert & McNair, 2012). Clarify how TMD was calculated.

Include your introductory citations here on the manuals used to calculate the data.

Line 205- remove ‘that’.

Line 207- missing ).

Procedure

Lines 210-212 is repetitive of detail already stated on lines192-195. Remove.

Some of the citations (lines 60-62) could be relocated to your methods section instead if this is where/when they were used.

Clarify why these checkpoint distances were selected over others. Also relate to checkpoint so it is clear which of the checkpoints were used e.g. (checkpoint 3) and (checkpoint ?). Also check distance of 68km checkpoint as current checkpoint 5 and 6 are 37 and 47.5 miles (59.5km, 47.5km respectively).

Detail reported in results (Lines 270-279) is better placed in the methods.

Regarding the exploratory analysis of ‘experience’ indicate the total n= this data relates to (re: line 272-273).

Lines 274-276 include standard deviation data. Consider reporting ranges.

The terminology or classification of ‘low experience’ is uncommon, perhaps something like novice or inexperienced ultra-runner would be clearer. Also ‘highly’ experienced is again presumptive, so perhaps ‘more’ experienced. Justify and define your choice of ‘experience’ and the use of the median split to determine these groups. Define what is meant by ‘highly experienced’ vs. ‘low experience’.

Statistical analysis

Missing details on how data were analysed (descriptives) and reported and information on statistical analysis (statistical tests, software etc). Report effect size calculation/type and reference scores for small, medium and large. Report 95% confidence intervals.

Results

Your introduction (lines 161-162) insinuates the association of mood fluctuations to performance but this needs associating with your study findings. Consider analysing data with a performance outcome reference, and relation of BRUMS measurement to performance.

Figure 1 title indicates TMD but this is not reported in this figure. Reword title for accuracy.

Figure 1 missing y-axis title.

Include statistical differences (*) on figures.

Lines 240-244- for consistency report effect size data.

Discussion

The discussion is fairly descriptive and reiterative of summarising the patterns seen in the result. Overall the depth and impact of your results are weakly interpreted and the overall discussion is lacking detail in terms of the experience-related data/outcomes. Furthermore, more discussion and depth around the impact and implications of these findings are needed. Consider the application of these findings and what relevance they may have to ultra-runner experience/success/performance etc.

Your abstract indicates the use of mood-regulation strategies during the event. What sort of strategies could you postulate and indicate for further research. You also indicate within tie points of the event, again using your data, indicate when/where or how it is best for athletes to identify as and when they could implement such strategies, what indicators could be used/monitored?

Consider the interpretation of your data profiles in accordance to the iceberg-type profile e.g. page 15, lines 303-304.

BRUMS approach was lacking anonymity after the baseline recording due to participants being verbally asked to complete the scales which brings an element of bias. Acknowledge and critique this within your discussion.

Line 296- should be ‘follow-up’.

Refer to relevant figures where necessary, e.g. line 300, line 304.

General sentence structuring is brief and short and the flow and sentences could be more concise on page 15.

Line 305- remove repetition of ‘(34km)’.

Line 306-307- You are self-citing a pre-print version of the same data contained in this work and do not agree this is a suitable approach as it is the same work, not necessarily a ‘previous study’ so be careful in this method. Perhaps you mean to refer specifically to the 110mile data? Otherwise you need to reframe this association to the pre-print citation as it is misleading.

Lines 307-309- please expand why this would be the biggest change. This whole paragraph needs better detail and explanation.

Line 311-312- include supportive citation or infer and indicate further understanding of or comparison in competitive to non-competitive scenario effects on negative emotions/tension are needed. Also, more depth and critique expected here as data suggests that ultra-endurance participation is not necessarily race competitive apart from the top runners for example, compared to this study whose highest ranking athlete was position 26. Most ultra-runners indicate a self-driven goal (completion, personal target/goal) for participant in such events, rather than per se race competition. This is where Howe et al., (2019) provide some benefit to establishing whether there is any competitive anxiety by removing the element of race and the goal is a self-driven completion.

Line 317 – ‘release-of-competitive-anxiety’ needs a citation.

Line 321- expand on the implications and relevance/ approaches of management pre-race arousal. Reword to ‘pre-race arousal’.

Lines 330-331- requires a citation.

Line 333-334- requires a citation.

Line 338- better to include citation to the two sample sizes for clarity.

Line 338-339- clarify if you mean with regards to the current study.

Lines 340-341- As identified previously, it is important to acknowledge the sample size were based on two events across two years and there may be external factors that could have impacted the mood states on the day of the race compared to the baseline (e.g. weather).

Line 343- proof read and check grammatical structure- needs rewording.

Line 351- acknowledge the potential for future research on these factors.

Rather than the summary, the results of the study need more expansion in the main discussion around the implications of these findings to performance and the sort of proposed modulation of psychological training by providing examples or indicating the need for future research (e.g. line 358).

Include critique of the work.

Based on the conclusions (lines 364-367) can you support this with your findings? Further analysis and comparison of the mood states with performance or success in the event would help to support or indicate if such mood fluctuations during an event have such performance effects, or whether adjustments are adapted to enable successful completion. More depth around the importance and effect your data shows is required. Would be useful to bring in more association and relevance perhaps to race pacing, performance and completion success (i.e. not to ‘DNF’ etc) and perhaps to associate between successful and unsuccessful race performance.

Within the main discussion more depth and association to the sort of strategies that could be implemented or have association to race success are needed.

Line 369-370- ‘resilience in other domains’ is vague and unclear. Reword this sentence context.

Additional

Line 373- define abbreviation ‘PB’ or state author ‘PB’ for clarity.

References

Citation list not ordered according to Vancouver style (journal guidelines). Order of citation list is also inaccurate.

Within-text citations need proofing and checking for accuracy, e.g. line 61 spelling does not match spelling in citation list. Citation Hanin, 2010 missing from list.

Replace Arent with more suitable citation relevant to study.

Remove Beedie et al., 2000 citation as not included.

Brick et al., 2018 conflicts with in-text citation Brick et al., 2020 (line 367)- correct and amend.

Rundfelt and Rundfeldt spelling mistakes- check and amend (in text /citation list).

Remove Lakeland 100 citation

Leunes needs amending to LeUnes

Remove Ultra Trail Du Mont Blanc. (2021) citation.

Reviewer #4: Introduction. A greater insight into psychological factors typical of the ultra-distance runner is missed:

Buck, K., Spittler, J., Reed, A., & Khodaee, M. (2018). Psychological attributes of ultramarathoners. Wilderness & environmental medicine, 29 (1), 66-71.

Roebuck, G. S., Urquhart, D. M., Che, X., Knox, L., Fitzgerald, P. B., Cicuttini, F. M., ... & Fitzgibbon, B. M. (2020). Psychological characteristics associated with ultra ‐ marathon running: An exploratory self ‐ report and psychophysiological study. Australian Journal of Psychology, 72 (3), 235-247.

Lane, A. M., & Wilson, M. (2011). Emotions and trait emotional intelligence among ultra-endurance runners. Journal of Science and Medicine in Sport, 14 (4), 358-362.

Participants: It would be interesting to compare the sample with others from similar studies.

Hoffman, M. D., Ong, J. C., & Wang, G. (2010). Historical analysis of participation in 161 km ultramarathons in North America. The International journal of the history of sport, 27 (11), 1877-1891.

Statistic analysis. The statistical treatment carried out is not detailed.

Data processing. Reliability is not shown.

The presentation of the results is not correct.

The conclusions drawn from the document may be attractive but it could have been interesting to compare results with the work of:

Lahart, I. M., Lane, A. M., Hulton, A., Williams, K., Godfrey, R., Pedlar, C., ... & Whyte, G. P. (2013). Challenges in maintaining emotion regulation in a sleep and energy deprived state induced by the 4800Km ultra-endurance bicycle race; The Race Across America (RAAM). Journal of sports science & medicine, 12 (3), 481.

Rochat, N., Hauw, D., Antonini Philippe, R., Crettaz von Roten, F., & Seifert, L. (2017). Comparison of vitality states of finishers and withdrawers in trail running: An enactive and phenomenological perspective. PLoS One, 12 (3), e0173667.

Does not relate the introduction and conclusions to important articles on the 2020 and 2021 theme

6. PLOS authors have the option to publish the peer review history of their article (what does this mean?). If published, this will include your full peer review and any attached files.

Reviewer #1: **Yes: **Andy Lane

Reviewer #2: No

Reviewer #3: No

Reviewer #4: No

---

## [Author Response · Author response to Decision Letter 0]

2 Jul 2021

Reviewer #1: 

An interesting study that examines mood state changes in ultra-endurance. Using the POMS model of mood, a series of hypotheses on how moods states will change are presented and tested. The author should focus on competitive mood throughout the article and so discussion of mood in exercise is out of scope of the study.

Response: We have redrafted the abstract and introduction to focus more narrowly on competitive mood and removed the references to exercise

The authors report mood state changes in treadmill studies. Whilst the distance is known, so is the terrain whereas in a real race, the terrain and environment conditions contribute to uncertainty. This connects to the notion that the authors should relate their findings to competitive settings. I believe these revisions are easily overcome.

Response: The reviewer makes a good point. Consistent with the approach of focusing more tightly on competition, we have rationalised our discussion of Howe et al., in the introduction, and now more explicitly draw the readers attention to the fact that Howe et al., describe a non-competitive event

The authors might wish to consider how they use the term mood, which are feelings which do not have a specific antecedent that is known to the participant, from emotion, where the cause of feelings is known (Beedie et al., 2010). It is possible for mood and emotion to change during an ultra.

Response: Thank you for raising this issue. We agree that this is an important distinction and have added a more explicit definition for the construct of ‘mood’, which clearly differentiates it from emotion (P2, line 42). As we cannot be certain whether or not participants were able to identify specific antecedents for their feelings we have used the term ‘mood’ throughout 

The researchers delimit the work to running. This is understandable. Yet, they have non-competitive running, which is also understandable. However, it made me wonder why they did not use research from ultra events that were not running, such as cycling. Is there a huge difference in mood states changes between ultra-endurance events? I am not sure, but given the exploratory aspect of this study, such proposals could be considered.

We agree that the study may have relevance for other ultra-endurance events such as cycling. However, given the strong steer from reviewer 2 to cut down the length of the manuscript we have decided not to discuss these other endurance events in detail. However, in order to alert the reader to the possible links with other ultra-events we have briefly mentioned how the pattern of results in ultramarathons is similar to that observed in other ultra events (p4, line 81-82)

Lane and Terry (2000) focused on depressed mood and argued that a no-depression – depression dichotomy was influential. Your data would allow the possibility to test whether an increase in depression, results in a much larger increase in negative mood. Whilst running will lead to an increase in fatigue, could depression accelerate perceptions of negative mood.

We agree that would certainly be an interesting question to answer. Unfortunately, the data probably don’t allow us to address this question. The main problem is that the depression scores were very low (M= 4.3, SD = 1.46), so the whole sample would seem to fall into ‘no depression’ group. Depression also did not increase in a statistically significant way across the different stages of the race. As a result we have not conducted further analysis, although the data are available on the OSF if the reviewer would be interested in exploring this issue further. 

Relevant references to consider

1. Beedie, C. J., Terry, P. C., & Lane, A. M. (2005). Distinguishing mood from emotion. Cognition and Emotion, 19, 847-878.

2. Lahart, I., Lane, A.M., Hulton, A., Williams, K., Charlesworth, S., Godfrey, R., Pedlar, C., George, K., Wilson, M., & Whyte, G., (2013). Challenges in maintaining emotion regulation in a sleep and energy deprived state induced by the 4800km ultra-endurance bicycle race; the Race Across America (RAAM). Journal of Sports Science and Medicine, 12, 481-488. http://www.jssm.org/vol12/n3/17/v12n3-17text.php

3. Lane, A. M., & Terry, P. C. (2000). The nature of mood: Development of a conceptual model with a focus on depression. Journal of Applied Sport Psychology, 12, 16-33.

4. Lane, A. M., Whyte, G. P., Shave, R., Barney, S., Stevens, M. J., & Wilson, M. (2005). Mood disturbance during cycling performance at extreme conditions. Journal of Sports Science and Medicine, 4, 52-57. http://www.jssm.org/vol4/n1/7/v4n1-7text.php

5. Pedlar, C. R., Lane, A. M., Lloyd, J. C., Dawson, J., Emegbo, S., Stanley, N., & Whyte, G. P. (2007). Sleep profiles and mood state changes during an expedition to the South Pole: a case study of a female explorer. Wilderness and Environmental Medicine, 18, 2, 127-132.

6. Stanley, D. M, Lane, A. M., Beedie, C. J., Devonport, T. J. (2012). “I run to feel better; so why I am thinking so negatively”. International Journal of Psychology and Behavioral Science, 2, 6, 28-213. 10.5923/j.ijpbs.20120206.03

Reviewer #2: 

Introduction:

• “the literature exploring the consequences of different types, durations and intensities of exercise on mood after bouts of exercise has reached a broad consensus that the optimal level of exercise for mood benefits in healthy adults is 20-30 minutes of moderate intensity exercise” -> Careful with this statement. There are also studies that show that higher intensities led to the highest increases of positive mood (Schmitt et al. 2020, Bartlett et al., 2011). It is not yet clear what role the fitness of the subjects plays when it comes to mood. In addition, many studies have often only examined moderate intensity instead of comparing different intensities.

We than the reviewer for bringing these studies to our attention. In order to address the comments of R1 and this reviewer regarding the length and focus on the Introduction we have redrafted the introduction to focus more explicitly on competitive ultrarunning. As a consequence the statement referred to here has been removed.

• double statement: “The study of mood states in ultrarunners has largely focused on mood immediately before and after an ultrarun, using the Profile of Mood States” and “Numerous studies have measured these dimensions before and after an ultramarathon and reported consistent findings.”

Deleted the second sentence

• The introduction is very long at should be shortened by half.

We have tried to address this point by removing some text and restructuring other paragraphs. We have not quite managed a 50% reduction in length, but we hope the reviewer will agree we have made considerable effort to make the introduction more concise.

Methods:

• Table 1: Abbreviations not explained (DNF, PW).

Corrected

• “All participants gave informed consent, and the study was approved by the Durham University Psychology Research Ethics Committee.” Already mentioned in “Participants” paragraph

Corrected 

• “Data were collected on the week prior to and during the 2018 and 2019 Hardmoors 60 races.” -> on the week prior and during is doubled in the following paragraph. Leave it out in this sentence -> Data were collected during the 2018 and 2019 Hardmoors 60 races.

Corrected

• The acquisition of the pre-race measurement “30 minutes and 1.5 hours prior to the race starting” gives potentially high variance as e.g. tension might be higher 30 min before race that 90 min before race. This large timeslot/lack of standardization should mentioned in the limitations.

We have added a paragraph to the discussion that considers some of the potential limitations of the study (p15, lines 323-331), including this issue. 

• You should think of performing a subgroup analysis with the five that did not complete the BRUMS for the Week Before measurement (but did complete all the race-day measurements) as one group without median split as additional information. Could be implemented as supplement.

We have included this analysis as a supplementary material, as requested. The results are similar to those we report in the main body of the paper, with the exception that the main effect for depression was no longer statistically significant. This we believe is because the main effect of depression was primarily driven by a dip in depression score on the day of the race, rather than a systematic increase in depression during the race itself. The data are also openly available on the OSF, so any interested reader may explore the data as the wish. 

Results:

• Why are only 30 subjects in the median split analysis? 26+5 should result in 31 subjects.

We thank the reviewer for drawing this to our attention. There was an error in the reporting of the exclusions. Of the 8 excluded participants 4 (not 5) were excluded because the failed to complete the week before measure and 2 (not 1) were excluded because they failed to complete all the within-race measurements. The text has been updated to correctly reflect the exclusions (P6, lines 131-134)

Minor:

• “The study was approved by the Durham University Dept of Psychology Research Ethics Committee” -> period at the end of the sentence is missing

• “which produce six subscales that:” -> delete “that”

• “Eight participants were removed from the analysis.” -> replace . by :

• “(Mood Factor: anger, confusion, depression, fatigue, tension, vigour). Repeated Measures ANOVA.” ->remove period after bracket: …vigour) Repeated Measures…

All corrected

General comments:

The manuscript contains unnecessary duplications and should be revised in this respect.

We have carefully proof-read the manuscript and corrected those duplications.

Reviewer #3: 

This article entitled “Mood fluctuations during a 103km ultramarathon” aimed to determine the impact of a single-day event continuous trail-ultramarathon on mood fluctuation. The key findings support that of previous studies in identifying pre-event vigour and reduced anger, depression and fatigue and as participants competed in the event, there was a decline in vigour and an increase in fatigue as expected. There were also increases in tension at the start of the event that declined as the event progressed.

However, it was not really clear the implications and association these changes in mood would have on performance success or participant adherence or motivations for example.

The work would have been strengthened with some comparative analysis to establish the impact of such patterns on either performance times, pacing, completion, drop-out/adherence, etc. The reader is left thinking, ‘so what’? Overall, the authors should consider the impact of why during race event mood fluctuations have importance?

We thank the reviewer for this valuable suggestion. To address this point we have added two analyses that attempts to relate mood fluctuations to the overall completion time. The first analysis examined the relationship between the Total Mood Disturbance score recorded at the pre-race measure with the completion time in minutes. There was no statistically significant relationship between these variables (page 10, lines 214-217). This is largely consistent with previous work showing that mood is poor predictor of performance in long-duration sports (e.g. Beedie et al., 2000). However, we also examined the relationship between the variability in TMD, which we measured using the SD of the TMS scores, and completion time (p10, lines 218-225). This analysis revealed a significant correlation between these variables, suggesting that fluctuations in mood are associated with poorer running times. We have revised the manuscript to give more prominence to these results.

This manuscript could be significantly improved with reworking the data and interpretation of the results with more robust analysis and interpretation in the discussion to consider the application of the findings.

We assume the reviewer is referring to the additional analysis suggested in the previous comment. This has been added to the results section and considered in the discussion

The methodology is clear however improvements to details can be made and include more clarification on the data analysis and statistical approaches rather than in the results (see reviewer suggestions).

General data analysis of Mood scores and statistical reporting is accurate but there are mistakes in the data type reporting (ordinal) and calculation of performance times in table 1. It is unclear to the reviewer why the authors used sub-analysis within their data to compare ‘low experience’ athletes and ‘high experience’ athletes and they do not clarify this analysis and methodological decision in the methods section.

This analysis was exploratory, in that we did not decide on this analysis until we had looked at the data and realised there was quite a range of experience. The rationale was simply that we were interested to see if experience made a difference. In the text we attempted to flag this to the reader with the statement “then conducted an exploratory analysis to examine whether experience in running ultramarathons was related to mood changes during the run”. However, to make it clear to the reader that this analysis was exploratory we have introduced a sub-heading to the results section (p11, line 230)

Data are deposited in the Open Science Framework and a pre-print version cited in sportrxiv. The authors would benefit from consider the scientific rigour and depth of the pre-print version to better reflect and improve the quality of the current submitted manuscript.

We are somewhat confused by this comment. The analysis in the manuscript is largely similar to that in the preprint. The data and methods are the same, so the level of scientific rigour in identical in the two studies. 

Generally the manuscript is appropriately formatted and presented however there are a number of grammatical, structural and presentation issues that can be reviewed to improve the clarity and accuracy of the work.

The authors cite and are aware of the relevant and seminal works of the topic area and critique and incorporate key studies linked to the work, however the reference list is not according got the Vancouver style and contains many errors and unused citations.

We thank the reviewer for pointing this out and have corrected this.

The authors should reconsider a robust reworking of the manuscript for accuracy and the analysis and interpretation of their findings with more application and are then encouraged to reconsider submitting a revised version with a more detailed critique and depth of the discussion particularly.

We have thoroughly revised the discussion as suggested to include sections explicitly relating the mood data to performance and noting the limitations of the study. 

Amendment suggestions:

Abstract

Line 15-16- the introductory line is unimpactful and not really relevant or necessary to the overall aim and focus of the work. Physical activity is to vague and sweeping to the relevance of ultra-endurance events. Consider rewording or expanding the implication towards ultra-endurance and why mood is of importance particularly during an event.

Line 23/ 31- It is not necessary or adding to the abstract to use the TUM abbreviation. I would remove.

done

Indicate the event is a single-day race.

Done

The terminology or classification of ‘low experience’ is uncommon, perhaps something like novice or inexperienced ultra-runner would be clearer. Also ‘highly’ experienced is again presumptive, so perhaps ‘more’ experienced.

Done

Reanalysis of the data will enable the summary to be more indicative of the implications and effect mood fluctuations DURING an event may play importance to runner success /performance for example.

We are grateful for this suggestion and have added an analysis, as suggested (p10, line 213-225)

Line 34- clarify what you mean by ‘in other domains’.

done 

Introduction

Line 41- replace citation ‘Arent et al., 2000’ with a more relevant citation to the study focus.

Line 46- you indicate and relate the importance of understanding of mood during long duration bouts and rationalise due to increases participation in ultra-endurance events but expand to clarify why this may be of importance. For example, is it likely to impact performance success etc.

Final sentence in the opening paragraph now explicitly argues that the issue is important with respect to optimising performance of athletes.

Not sure your abbreviations TUM and TUR are necessary.

Line 50- update this data and citation with the more recent dataset that is available.

Line 53 – add supportive citation(s).

Line 58- ‘The study’ is grammatically incorrect, check sentence structure.

These sections have been removed from the manuscript to reduce the length of the intro and keep the focus tightly on competitive running, as requested by R1 and R2

Lines 58-62. In addition to the citations that support the tools of assessing mood, the sentence should also be supported with citations of at least example studies where these tools have been specifically used in ultra-runner studies. You could consider removing unnecessary citations and only citing the seminal citations (.e.g McNair et al., 1971; 1992) for the scale development. 

Done

Also indicate the scale (POMS) is used to assess transient mood states. Some of these citations could be relocated to your methods section instead.

Done

Line 64- Clarify that TMD is a global indicator and in methods indicate how this is calculated.

Done

Lines 65-66 is vague and unsupported in depth.

Line 65- ‘Numerous studies have…’ should be supported by citations, and identify the seminal or key examples as more than one should support this claim ‘numerous’. Line 66- ‘reported consistent findings’, you need to expand this claim/statement with what exactly they report or identify as key outcomes.

Line 66-68 is supported with an irrelevant citation and the sentence claims inaccurate reflections of ultra-marathon requirements. Replace ‘Baron et al., 2009’ with a more suitable citation.

Line 68 – ‘perception of physiological resources’ needs better clarification and detail around what this means, particularly in relation to ultra-endurance performance.

The lines here are not central to our overall line of argument, so in the interest of reducing MS length we have removed lines 65-68

Line 75-80 – the grammatical structure and clarity of the sentences needs improving. It is unclear what is being stated. Furthermore, this needs more clarification/explanation to why in-event monitoring would be beneficial to Weiner’s position and also how it relates to ultra-marathon performance/success/outcomes.

In the interest of reducing the length of the introduction this material has been removed 

Line 81-83 - Vague sentence start. Instead of claiming ‘others’ cite who those ‘others’ are. Furthermore the sentence is unclear and needs rewording and checking the grammatical structure.

Sentence has been revised to make it clear that the others referred to were Lane, Robinson & Clark 2019

Line 83-85- do athletes ‘begin’ or have available to them to implement strategies and what sort of strategies are used. Expand clarity and detail here.

Sentences removed.

Line 85- Hanin, 2010 citation missing from reference list.

Citation has been removed from text 

Line 85- ‘…few studies’ support or indicate what studies (citations) do exist to support this statement.

Rephrased to make it clear the sentence is referring to the 3 studies cited in the previous paragraph 

Line 85- again support / expand to explain why a baseline compared to pre-race measure of emotion would be beneficial. 

Line 86-89 – expand clarity to why a within event indication of ‘tension’ would be useful. Elaborate on the Lane et al., explanation (and include citation date), and ‘Goal completion account’ needs a supportive citation and elaboration.

Line 89- ‘…or follows some other profile’. What is meant by this, expand and give examples of such possible profiling.

This section has been redrafted to more precisely explain the differing predictions made by the explanations put forward by Rauch et al., (1988) and Lane et al., (2019) (p3, lines 55-64)

Line 90- ‘anxiolytic effect’. Also expand what is meant by this term.

Anxiolytic refers to an intervention or medication that reduces anxiety. We have replaced this term with the plain English phrase ‘anxiety reducing’ 

Line 92- ‘….produce negative’, check grammatical sentence structure.

Checked, the structure is fine

Also be critical as ‘Reed and Ones, 2006 is limited in interpretation and ultra—distance events are very different to acute shorter duration bouts.

Overall from this point, the introduction needs more depth of the importance of all this detail and the emotion and profiling to ultra-marathon running and performance, such as success.

The revised introduction now offers a more nuanced discussion of the relationship between mood and performance, noting that pre-race mood scores are poor predictors of performance but that variability in mood scores may be more useful, as suggested by the reviewer.

Line 96- ‘Several studies’ again this claim needs citation support.

Citations added

Line 100- Clarify the abbreviation ‘BRUMS’ and include the seminal citation (Terry et al., 1999; 2003) for this short version scale, and also explain it is a short version of POMS etc. Move the detail cited on page 7 lines 127-128 to here.

Detail added to p7, lines 147-154

Line 101- ‘Consistent with previous work…’ support this claim with suitable citations.

Phrase has been removed during the redrafting

Lines 102-104- ‘Confusion and anger changed’ was quite vague and unclear what sort of change and why this was of importance. Also include citation for TEI (Petrides, 2001). Expand briefly what TEI represents/indicates.

Redrafted as requested

Line 105- ‘1st’ – write as ‘first’.

Done

Line 105-106 is unclear. What is meant when you say ‘consistent with a release-of-anxiety explanation…’ Check clarity and amend this sentence.

Rephrased to “these data suggest that changes in tension in multi day ultramarathons are best explained in terms of a reduction of anxiety that occurs at the start of races.”

Line 106-Punctuation- ‘..’ at end of sentence.

Corrected 

Line 108- what is ‘non’? Better terminology use needed here.

Have revised to hyphenate ‘non-competitive’, meaning a long distance run that is not a competition. 

Line 108- spelling mistake ‘sub-arctic’.

Line 109- degrees ‘Celsius’.

Considering the detail, for example at Lines 110-onwards. To help your result explanation and interpretation perhaps needs to relate to Morgan’s Mental Health Model (1980;1985) iceberg-type profile, to help better clarity in explaining the effect/pattern of data seen.

Line 111- spelling ‘in in’

Line 112- grammar of sentence ‘in similar’.

Line 115- Rausch Goal Attainment needs citation.

Line 116-‘ anxiety account’ is unclear and needs clarification of what is meant by this sentence detail.

Line 116-117- ‘similarly harsh’ – appreciate the implication here, but also be critical of the fact its one temperature extreme to another.

Line 117- ‘to the other multi-day studies’. Remove ‘other’ and also support with example citations.

Lines 116-119- short sentence use. Section can be more concisely written.

Line 121- ‘the impact of physical activity’ I think better to term ‘physical exertion’ is better refection of the efforts of ultra-endurance events.

From this point, again there needs expansion to why during race event mood states are of importance or what sort of impact/relevance it is likely to have.

Line 140- grammar ‘will’ to ‘would’.

Line 141-143- Citation needed to support sentence.

Line 146- ‘decrease’ not ‘decreases’.

Line 148- ‘a significant improvement’.

Line 154- ‘first’ not ‘1st’.

Your critique on lines 129-157 – consider the impact of the data.

Line 160 ‘ measurements’.

Lines 161-162 contradicts the subsequent lines 163-164. So make it clear where the differences or lack of evidence lies and clarify what are the novel aspects to your research and why this is of importance/relevance.

We have removed or substantially revised lines 108-162

Methods

Race details

Indicate the event is a single-day race with an 18 hour time restriction to complete the distance with check-point cut-off times.

Done

Line 179- define abbreviation (HM60)

Line 180- ‘The Association of Running Clubs’ (2021). Check grammar or amend citation list detail

Lines 181-183- remove this detail as does not add to the focus and interpretation of the work.

Lines 183- 185- include citation/source for detail.

All corrected 

Participants

The detail provided in results (lines 224-226 etc.) is better reported in the methods. 

We have moved this material to the Participants section, p 6 lines 127-134

Again proofread to remove short sentence use and be more concise.

Line 195- full stop missing.

Corrected 

Lines 191-192- Considering two races across two years were used to recruit participants, it is importance to acknowledge the race conditions for these two years (and whether or Not race checkpoints/regulations were the same) as the weather for example may impact the mood states during the events and therefore is a potential limitation to the grouping of your data and mood fluctuations. Also acknowledge and identify this critique in your discussion.

A limitations section has been added to the discussion in which we consider this critique (p15 lines 324-332)

Table 1 (page 11).

Need to footnote and define what DNF abbreviation is.

Done

Include units of measure for each variable.

done

Also report n= numbers for example, did all participants in the data set complete ultras over 80km or 160km- it may be more informative to report of the cohort the n= or % (frequency).

Again for DNFs’ it may be better to report of cohort, how many reported DNFs and n=events (frequency).

Done

Regarding race position, did all participants complete /finish and place.

Your race position ordinal numbers cited in the table need to be rounded to whole numbers – you cannot finish in 118.19th place

Done

Check and reanalysis your finish time data (reported as mins) as this is not accurate. 96 mins to complete 103kms!?

This is a typo and has been corrected to 960. 

Materials

More depth/detail of the scale needed. Indicate the short-version form of the POMS was used. Clarify the six scales calculated and the TMD (Heuchert & McNair, 2012). Clarify how TMD was calculated.

We used the Brunel Mood Scale and clarified that TMD was calculated as the sum of the negative mood scores (anger, confusion, depression, fatigue, tension) minus vigour. This detail has been added to p7, line 152

Include your introductory citations here on the manuals used to calculate the data.

Line 205- remove ‘that’.

done

Line 207- missing ).

Corrected 

Procedure

Lines 210-212 is repetitive of detail already stated on lines192-195. Remove.

Done

Some of the citations (lines 60-62) could be relocated to your methods section instead if this is where/when they were used.

Done

Clarify why these checkpoint distances were selected over others. Also relate to checkpoint so it is clear which of the checkpoints were used e.g. (checkpoint 3) and (checkpoint ?). Also check distance of 68km checkpoint as current checkpoint 5 and 6 are 37 and 47.5 miles (59.5km, 47.5km respectively).

There were several pragmatic reasons for selecting checkpoint 3 and 6 for the within race measurements:

1. The spacings between the checkpoints were selected to reduce the amount of occasions in which we would have runners across multiple checkpoints, at worst requiring members of the team to be at 2 checkpoints at the same time.

2. At both checkpoints runners had drop bags meaning most were highly likely to be stopping for longer periods at these points. This was not the case at other checkpoints, meaning that data collection would have less of an impact on participant’s race time. 

3. Parking considerations and access needed to be considered, the selected sites had good parking and at Ravenscar(checkpoint 6) we had permission to park outside the hall and be based in the hall, whereas no other crew were allowed at this point. This was helpful for identifying runners. The North Yorkshire coast in summer is incredibly busy and challenging in respects to this at other CP's. 

We also thank the reviewer for picking up on the discrepancy between the route description we cite and the text of the manuscript. We have recalculated the conversion from miles to km and now report the correct distances in the manuscript. Checkpoint 3 is at 35.4km and checkpoint 6 is at 66km from the start. 

Detail reported in results (Lines 270-279) is better placed in the methods.

Regarding the exploratory analysis of ‘experience’ indicate the total n= this data relates to (re: line 272-273).

Added

Lines 274-276 include standard deviation data. Consider reporting ranges.

Included

The terminology or classification of ‘low experience’ is uncommon, perhaps something like novice or inexperienced ultra-runner would be clearer. Also ‘highly’ experienced is again presumptive, so perhaps ‘more’ experienced. Justify and define your choice of ‘experience’ and the use of the median split to determine these groups. Define what is meant by ‘highly experienced’ vs. ‘low experience’.

We have adjusted the terminology to refer to ‘less’ and ‘more’ experienced groups to make it clear to the reader that this is a relative, rather than absolute categorisation. As the reviewer notes, there are no broadly agreed method of classifying athletes as low or high experience, so we believe a median split offers a reliable method of creating two groups such that one group has relatively more experience than the other. 

Statistical analysis

Missing details on how data were analysed (descriptives) and reported and information on statistical analysis (statistical tests, software etc). Report effect size calculation/type and reference scores for small, medium and large. Report 95% confidence intervals.

We have added details on the software at the start of the Results section. Effect sizes are now reported throughout (partial eta squared for ANOVAs and Cohen’s D for t-tests). 95% confidence intervals are reported on the figures. 

Results

Your introduction (lines 161-162) insinuates the association of mood fluctuations to performance but this needs associating with your study findings. Consider analysing data with a performance outcome reference, and relation of BRUMS measurement to performance.

This analysis has been added to p10, lines 215-225

Figure 1 title indicates TMD but this is not reported in this figure. Reword title for accuracy.

Completed 

Figure 1 missing y-axis title.

Include statistical differences (*) on figures.

Corrected

Lines 240-244- for consistency report effect size data.

Effect sizes added

Discussion

The discussion is fairly descriptive and reiterative of summarising the patterns seen in the result. Overall the depth and impact of your results are weakly interpreted and the overall discussion is lacking detail in terms of the experience-related data/outcomes. Furthermore, more discussion and depth around the impact and implications of these findings are needed. Consider the application of these findings and what relevance they may have to ultra-runner experience/success/performance etc.

We have revised the discussion to more briefly summarize the key finds and interpret the performance data in terms of Marcora’s Psychobiological Model (PBM; Marcora, 2017) (p13, lines 262-278)

Your abstract indicates the use of mood-regulation strategies during the event. What sort of strategies could you postulate and indicate for further research. You also indicate within tie points of the event, again using your data, indicate when/where or how it is best for athletes to identify as and when they could implement such strategies, what indicators could be used/monitored?

We have revised the abstract to remove reference to mood regulation strategies.

Consider the interpretation of your data profiles in accordance to the iceberg-type profile e.g. page 15, lines 303-304.

As we understand it, the iceberg profile refers to a comparison between athletes and a normal population. As we do not seek to compare groups, but rather make a within-participants comparison of mood over time it is not immediately clear to us how the inverted v shaped profile for TMD relates to the iceberg profile, other than a superficial similarity in shape. We have therefore decided not to include this comparison in the discussion.

BRUMS approach was lacking anonymity after the baseline recording due to participants being verbally asked to complete the scales which brings an element of bias. Acknowledge and critique this within your discussion.

We now consider this issue in a paragraph on limitations

Line 296- should be ‘follow-up’.

Corrected

Refer to relevant figures where necessary, e.g. line 300, line 304.

General sentence structuring is brief and short and the flow and sentences could be more concise on page 15.

Line 305- remove repetition of ‘(34km)’.

Completed 

Line 306-307- You are self-citing a pre-print version of the same data contained in this work and do not agree this is a suitable approach as it is the same work, not necessarily a ‘previous study’ so be careful in this method. Perhaps you mean to refer specifically to the 110mile data? Otherwise you need to reframe this association to the pre-print citation as it is misleading.

We have removed reference to the preprint

Lines 307-309- please expand why this would be the biggest change. This whole paragraph needs better detail and explanation.

The primary goal of ultrarunners is typically to complete the race (they typically score very highly on mastery achievement orientation and, as the reviewer notes in the next item, tend to report setting self-referential completion goals). The athlete cannot know whether this goal will be achieved until the end of the race. If Tharion is correct, and the reduction in tension is caused by the achievement of the goal of completing the race, it follows that the reduction in tension should be observed at the end of the race, when the goal has been completed. We have revised this line of argument to make this more explicit to the reader (p283-297)

Line 311-312- include supportive citation or infer and indicate further understanding of or comparison in competitive to non-competitive scenario effects on negative emotions/tension are needed. Also, more depth and critique expected here as data suggests that ultra-endurance participation is not necessarily race competitive apart from the top runners for example, compared to this study whose highest ranking athlete was position 26. Most ultra-runners indicate a self-driven goal (completion, personal target/goal) for participant in such events, rather than per se race competition. This is where Howe et al., (2019) provide some benefit to establishing whether there is any competitive anxiety by removing the element of race and the goal is a self-driven completion.

We have revised this paragraph to consider the motivational goals of the ultrarunners (p14, line 286-293)

Line 317 – ‘release-of-competitive-anxiety’ needs a citation.

Citation to Lane et al., (2018) added 

Line 321- expand on the implications and relevance/ approaches of management pre-race arousal. Reword to ‘pre-race arousal’.

This paragraph has been rewritten to offer a more balanced interpretation of the data. Given that we found no evidence that pre-race mood predicted performance we have removed reference to the manipulation of pre-race mood being of potential interest and focused the paragraph on explaining the possible mechanism for the effect we observed.

Line 338- better to include citation to the two sample sizes for clarity.

Line 338-339- clarify if you mean with regards to the current study.

Clarified 

Lines 340-341- As identified previously, it is important to acknowledge the sample size were based on two events across two years and there may be external factors that could have impacted the mood states on the day of the race compared to the baseline (e.g. weather).

This is an important point, and we now bring this to the attention of the reader in the ‘limitations’ section. We also conducted some post-hoc t-tests to check whether there were any statistically significant differences in mood. There were no differences (all t values .85 or lower, all P values .41 or higher) which we hope will help convince the reviewer that this aspect of the design is not responsible for the results. 

Line 343- proof read and check grammatical structure- needs rewording.

Completed 

Line 351- acknowledge the potential for future research on these factors.

Added the following text to the discussion (line 368)

“Indeed, the role of these factors on mood are not well understood and may be a fruitful avenue for future research.”

Rather than the summary, the results of the study need more expansion in the main discussion around the implications of these findings to performance and the sort of proposed modulation of psychological training by providing examples or indicating the need for future research (e.g. line 358).

Include critique of the work.

Based on the conclusions (lines 364-367) can you support this with your findings? Further analysis and comparison of the mood states with performance or success in the event would help to support or indicate if such mood fluctuations during an event have such performance effects, or whether adjustments are adapted to enable successful completion. More depth around the importance and effect your data shows is required. Would be useful to bring in more association and relevance perhaps to race pacing, performance and completion success (i.e. not to ‘DNF’ etc) and perhaps to associate between successful and unsuccessful race performance.

Within the main discussion more depth and association to the sort of strategies that could be implemented or have association to race success are needed.

We have rewritten the concluding paragraph to more accurately reflect the data and removed the more speculative conclusions about interventions

Line 369-370- ‘resilience in other domains’ is vague and unclear. Reword this sentence context.

Rephrased as requested

Additional

Line 373- define abbreviation ‘PB’ or state author ‘PB’ for clarity.

Done

References

Citation list not ordered according to Vancouver style (journal guidelines). Order of citation list is also inaccurate.

Within-text citations need proofing and checking for accuracy, e.g. line 61 spelling does not match spelling in citation list. Citation Hanin, 2010 missing from list.

Replace Arent with more suitable citation relevant to study.

Remove Beedie et al., 2000 citation as not included.

Brick et al., 2018 conflicts with in-text citation Brick et al., 2020 (line 367)- correct and amend.

Rundfelt and Rundfeldt spelling mistakes- check and amend (in text /citation list).

Remove Lakeland 100 citation

Leunes needs amending to LeUnes

Remove Ultra Trail Du Mont Blanc. (2021) citation.

References corrected 

Reviewer #4: 

Introduction. A greater insight into psychological factors typical of the ultra-distance runner is missed:

Buck, K., Spittler, J., Reed, A., & Khodaee, M. (2018). Psychological attributes of ultramarathoners. Wilderness & environmental medicine, 29 (1), 66-71.

Roebuck, G. S., Urquhart, D. M., Che, X., Knox, L., Fitzgerald, P. B., Cicuttini, F. M., ... & Fitzgibbon, B. M. (2020). Psychological characteristics associated with ultra ‐ marathon running: An exploratory self ‐ report and psychophysiological study. Australian Journal of Psychology, 72 (3), 235-247.

Lane, A. M., & Wilson, M. (2011). Emotions and trait emotional intelligence among ultra-endurance runners. Journal of Science and Medicine in Sport, 14 (4), 358-362.

We agree that the psychology of ultra-distance runners is a topic of great interest. However, the primary goal of the study was to explore mood rather than personality per se, and in the interest of presenting a concise, clear and coherent rationale for the study we have decided to retain our original focus on mood. 

Participants: It would be interesting to compare the sample with others from similar studies.

Hoffman, M. D., Ong, J. C., & Wang, G. (2010). Historical analysis of participation in 161 km ultramarathons in North America. The International journal of the history of sport, 27 (11), 1877-1891.

This would indeed be interesting, but in our view this would be hard to justify with respect to the scientific rationale for our study. The data are available on the OSF, and the reviewer is welcome to use them to explore this comparison for themselves. 

Statistic analysis. The statistical treatment carried out is not detailed.

We have revised the analysis section to make the use of ANOVA, t-tests and pearson correlations explicit

Data processing. Reliability is not shown.

We do not understand this comment. Perhaps the reviewer could be more specific about what measures they would like tested for reliability. 

The presentation of the results is not correct.

We have revised the results section, but without more specific feedback from the reviewer it is hard to know which aspects of the results the reviewer would like us to change. 

The conclusions drawn from the document may be attractive but it could have been interesting to compare results with the work of:

Lahart, I. M., Lane, A. M., Hulton, A., Williams, K., Godfrey, R., Pedlar, C., ... & Whyte, G. P. (2013). Challenges in maintaining emotion regulation in a sleep and energy deprived state induced by the 4800Km ultra-endurance bicycle race; The Race Across America (RAAM). Journal of sports science & medicine, 12 (3), 481.

This is an interesting study that shows sleep deprivation leads to reduced ability to regulate emotion. The comparison to the current study is a bit hard to make as although the participants were awake for a long time, they were not sleep deprived per se.

Rochat, N., Hauw, D., Antonini Philippe, R., Crettaz von Roten, F., & Seifert, L. (2017). Comparison of vitality states of finishers and withdrawers in trail running: An enactive and phenomenological perspective. PLoS One, 12 (3), e0173667.

Unfortunately only two of the participants failed to complete the race, so the sample size is too small to meaningfully compare finishers and non-finishers in our study 

Does not relate the introduction and conclusions to important articles on the 2020 and 2021 theme

We do not understand what this comment refers to. What is the 2020 and 2021 theme? Perhaps the reviewer could be more specific about which articles they are referring to.

---

## [Decision Letter · Decision Letter 1]

9 Aug 2021

PONE-D-21-06919R1

Mood fluctuations are associated with performance during ultrarunnning

PLOS ONE

Dear Dr. Smith,

Thank you for submitting your manuscript to PLOS ONE. After careful consideration, we feel that it has merit but does not fully meet PLOS ONE’s publication criteria as it currently stands. Therefore, we invite you to submit a revised version of the manuscript that addresses the points raised during the review process.

The following comments should be addressed:

Revise reference inserts format when more than two manuscripts are included. For example, in line 60-61, replace [1,2,3] to [1-3], or in line 78 replace to [11-18].Lines 197-199. Please include a proper “Statistical analysis” subsection, including **all** information regarding any statistical analysis performed, software used, etc., at the end of the “Methods” section. Information such as the one reported in lines 203-204 should be also included in this section rather than in the Results section.Lines 144-145. While this comment is adequate, “kilometers” have appeared in the manuscript from the abstract. Therefore, the full name or the abbreviation of kilometers could be removed from these lines as this unit has been widely used previously in the manuscript.Decimal places should be carefully revised. In my opinion one decimal place for age is enough and more adequate.  Furthermore, the same decimal places in the mean and in the SD should be included for the same parameter (for example in Table 1, there’s no sense to include two decimals for the SD of parameters with no decimal places in the mean). In addition if the measure does not allow any decimal places they should not be used.Table 1. Parameters and units should be clarified. For example: Previous Ultra's over 80km (n). The use of “PW” could be avoided as the complete form “per week” could be included. Actually, the term “training level (km/week” could be used, with a proper explanation in the foot-note of the table.Lines 262-278. Describe this exploratory analysis previously, in the Methods section (statistical analysis), and report only the results in this “Results” section.Figure legends. Show the symbol used rather than descriptions such as “stars” or “white discs”.Uniformize expressions such as 49km (for example in line 269) and 35.4 km (line 289). In my opinion a space should be left between the number and the unit (49 km, 35.4 km).Conclusion section should be shortened without loosing significance. For example, the first sentence can be removed as it is such as the fourth time the aim of the study is stated (abstract, introduction, beginning of the discussion and conclusions). Comparisons with other studies can be also removed as they are discussed properly in the discussion section (or instead, moved to the discussion section).

We look forward to receiving your revised manuscript.

Kind regards,

Pedro Tauler, Ph.D.

Academic Editor

PLOS ONE

Journal Requirements:

Additional Editor Comments (if provided):

Reviewers' comments:

Reviewer's Responses to Questions

**Comments to the Author**

1. If the authors have adequately addressed your comments raised in a previous round of review and you feel that this manuscript is now acceptable for publication, you may indicate that here to bypass the “Comments to the Author” section, enter your conflict of interest statement in the “Confidential to Editor” section, and submit your "Accept" recommendation.

Reviewer #2: All comments have been addressed

2. Is the manuscript technically sound, and do the data support the conclusions?

Reviewer #2: Yes

3. Has the statistical analysis been performed appropriately and rigorously? 

Reviewer #2: Yes

4. Have the authors made all data underlying the findings in their manuscript fully available?

Reviewer #2: Yes

5. Is the manuscript presented in an intelligible fashion and written in standard English?

Reviewer #2: Yes

6. Review Comments to the Author

Reviewer #2: (No Response)

7. PLOS authors have the option to publish the peer review history of their article (what does this mean?). If published, this will include your full peer review and any attached files.

Reviewer #2: No

---

## [Author Response · Author response to Decision Letter 1]

16 Aug 2021

The following comments should be addressed:

• Revise reference inserts format when more than two manuscripts are included. For example, in line 60-61, replace [1,2,3] to [1-3], or in line 78 replace to [11-18].

References have been amended accordingly. We have not changed the reference on line 84 because the citations are non-consecutive

• Lines 197-199. Please include a proper “Statistical analysis” subsection, including all information regarding any statistical analysis performed, software used, etc., at the end of the “Methods” section. Information such as the one reported in lines 203-204 should be also included in this section rather than in the Results section.

We have included a Statistical analysis section and moved all the details of the analyses to this section. 

• Lines 144-145. While this comment is adequate, “kilometers” have appeared in the manuscript from the abstract. Therefore, the full name or the abbreviation of kilometers could be removed from these lines as this unit has been widely used previously in the manuscript.

• Decimal places should be carefully revised. In my opinion one decimal place for age is enough and more adequate. Furthermore, the same decimal places in the mean and in the SD should be included for the same parameter (for example in Table 1, there’s no sense to include two decimals for the SD of parameters with no decimal places in the mean). In addition if the measure does not allow any decimal places they should not be used.

Decimal places have been revised as suggested 

• Table 1. Parameters and units should be clarified. For example: Previous Ultra's over 80km (n). The use of “PW” could be avoided as the complete form “per week” could be included. Actually, the term “training level (km/week” could be used, with a proper explanation in the foot-note of the table.

Parameter names have been revised to make them more informative and appropriate footnotes added

• Lines 262-278. Describe this exploratory analysis previously, in the Methods section (statistical analysis), and report only the results in this “Results” section.

This section has been moved, as required

• Figure legends. Show the symbol used rather than descriptions such as “stars” or “white discs”.

Legends revised 

• Uniformize expressions such as 49km (for example in line 269) and 35.4 km (line 289). In my opinion a space should be left between the number and the unit (49 km, 35.4 km).

The manuscript has been revised to ensure a space has been left between numbers and units.

• Conclusion section should be shortened without loosing significance. For example, the first sentence can be removed as it is such as the fourth time the aim of the study is stated (abstract, introduction, beginning of the discussion and conclusions). Comparisons with other studies can be also removed as they are discussed properly in the discussion section (or instead, moved to the discussion section).

The conclusion has been revised as suggested make it more concise.

---

## [Editor Report · Decision Letter 2]

18 Aug 2021

Reduced mood variability is associated with enhanced performance during ultrarunnning

PONE-D-21-06919R2

Dear Dr. Smith,

We’re pleased to inform you that your manuscript has been judged scientifically suitable for publication and will be formally accepted for publication once it meets all outstanding technical requirements.

Kind regards,

Pedro Tauler, Ph.D.

Academic Editor

PLOS ONE
---

## [Editor Report · Acceptance letter]

20 Aug 2021

PONE-D-21-06919R2 

Reduced mood variability is associated with enhanced performance during ultrarunnning 

Dear Dr. Smith:

I'm pleased to inform you that your manuscript has been deemed suitable for publication in PLOS ONE. Congratulations! Your manuscript is now with our production department. 

Kind regards, 

on behalf of

Dr. Pedro Tauler 

Academic Editor

PLOS ONE